# A Novel Low-Frequency Piezoelectric Motor Modulated by an Electromagnetic Field

**Jichun Xing \***  **and Yong Qin**

School of Mechanical Engineering, Yanshan University, Qinhuangdao 066004, China; qy1993@stumail.ysu.edu.cn
\* Correspondence: xingjichun@ysu.edu.cn

**Abstract:** For expanding the driving mode of the piezoelectric motor, a novel piezoelectric motor modulated by a magnetic field is proposed. This driving system combines piezoelectric driving and magnetic modulation together and can transform the reciprocating swing of the stator into step running of the rotor via the intermittent magnetic clamping between the rotor and stator. For investigating the inherent character of dynamics, the dynamic equations of key parts of the driving system are established. The natural frequencies and mode functions of the driving system are solved. A prototype was fabricated to prove the dynamic analysis and measure the output characteristic. The results show that the nature of the frequency measured from the test is coincident with theoretical analysis. In addition, by applying the driving frequency of 3 Hz, the voltage of the modulating signal of 4.5 V, the phase difference $\alpha$ between driving signal and modulating signal of 30°, the ideal outputs are 0.1046 rad/min for velocity and 0.405 Nmm for torque.

**Keywords:** electromagnetic modulation; piezoelectric motor; modal function; output characteristic

## 1. Introduction

With the continuous development of modern technology, the requirements for driving devices in intelligent manufacturing, optical fiber docking, biomedicine, and focusing systems are increasing. Piezoelectric motors are widely used in these fields because of their high displacement resolution, simple structure, and fast response [1,2]. For example, piezoelectric microrobots can be used in surgery, human digestive system monitoring and drug delivery [3,4]. Although many piezoelectric motors have been designed for various purposes, they can be roughly divided into three categories in terms of the driving principle: ultrasonic motors, inchworm piezoelectric motors and inertia piezoelectric motors [5].

The ultrasonic piezoelectric motor is the earliest and most widely used piezoelectric motor. According to the different driving signal waveforms, it can be divided into traveling wave ultrasonic motors and standing wave ultrasonic motors. Markus K. et al. [6], proposed a new type of traveling wave ultrasonic motor. The double input sliding mode controller, which uses frequency and phase difference as instructions, switches the control domain without interrupting the speed, reduces the control amount of phase difference, and does not produce overshoot. Chen W. et al. [7], proposed a radial bending ring traveling wave ultrasonic motor. The 40 slots on the outer surface of the stator ring are alternately filled with 20 piezoelectric stacks and 20 elastic blocks for generating traveling waves. The maximum speed of the motor can reach 146 r/min and the maximum torque is 1 Nm. Izuhara S. et al. [8], proposed a piezoelectric ultrasonic linear motor with hollow rectangular stator. The motor is a standing wave ultrasonic motor, which can directly drive its internal load through the stator. The structure saves the mechanism space, and has the advantages of fast response speed and high resolution.

The inchworm piezoelectric motor is a precision driving device that imitates the movement state of the inchworm and uses piezoelectric actuators. Mohammad T. et al. [9], proposed a two-axis inchworm piezoelectric motor, which consists of two integrated stages. Each stage provides continuous motion on one axis. It can be applied to an integrated precision positioning system with large stroke and multi-axis linkage. Xing J. et al. [10], proposed a new type of rotary inchworm piezoelectric motor. Due to the special structure of the stator, the driving mechanism can produce angular displacement during operation. The inchworm motor has the advantages of realizing adjustable clamping devices, and a simple structure of single parts. Dong H. et al. [11], proposed a linear inchworm piezoelectric motor, which consists of a driving mechanism and two clamping mechanisms. The overall structure of the motor is relatively simple, the power consumption is low, and the maximum operating speed is 0.72 mm/s.

An inertial piezoelectric motor is a motor that uses inertial impact force or torque to obtain linear or rotary motion. By varying the applied voltage, the actuators can take any position and therefore achieve the high motion resolution that is typical for piezo ceramics. Andrius C. et al. [12], designed a new type of inertial piezoelectric motor, consisting of a rectangular bronze stator, a clamping frame, a piezoelectric plate, and a conical rotor. The motor has a simple structure, good motion stability, and high space utilization. Wang L. et al. [13], proposed a new type of piezoelectric inertial rotary motor, which can be used to drive underwater micro-robots. The motor is composed of two rotors and a disc stator. It has a simple structure, small size, and good waterproof performance. The emergence of this motor has opened up a new way to develop advanced and flexible multifunctional robots. Mazeika D. et al. [14], designed a 3-degree-of-freedom high-resolution inertial piezoelectric motor. The motor can work in two vibration modes, namely the bending vibration mode of the outer shaft of the disc and the radial vibration mode of the disc. The bending vibration mode is used to rotate the rotor around the $x$ and $y$-axis. The radial vibration mode is used to rotate the rotor around the $z$-axis. Therefore, 3 degrees of freedom movement of the motor is realized.

In summary, although piezoelectric motors have many advantages compared with traditional electromagnetic motors, the driving pattern of these piezoelectric motors is based on rigid contact between the fixed rotors to transmit power. The long-term effects of friction and preload force will cause damage to the runner or stator and even lead to the failure of the operation. Therefore, controlling the preload to reduce the friction between the contact surfaces is a key issue [15,16]. Introducing electromagnetic force into the piezoelectric motor improves not only the controllability of the clamping force or preload but also the friction and wear between the stator and rotor are effectively reduced. The proposed piezoelectric motor expands the topology structure of piezoelectric motors, and combining the piezoelectric drive and electromagnetic modulation into one motor would enrich the driving principle of the piezoelectric motor. In this paper, we mainly established the dynamics equations of key parts and solved natural frequencies and mode functions of driving system. We also conduct an experiment to measure the output characteristic of a prototype.

## 2. Motor Structure and Working Principle

### 2.1. Motor Structure Composition

The proposed piezoelectric motor is mainly composed of the driving part, the electromagnetic modulation part and the moving part. The driving part includes two piezoelectric stacks, a flexible amplifying mechanism and an intermediate shaft. The electromagnetic modulation part includes an electromagnet, a disc-shaped slider, and a spring. The rotor shaft and a bearing belong to the moving part of the motor, as shown in Figure 1.

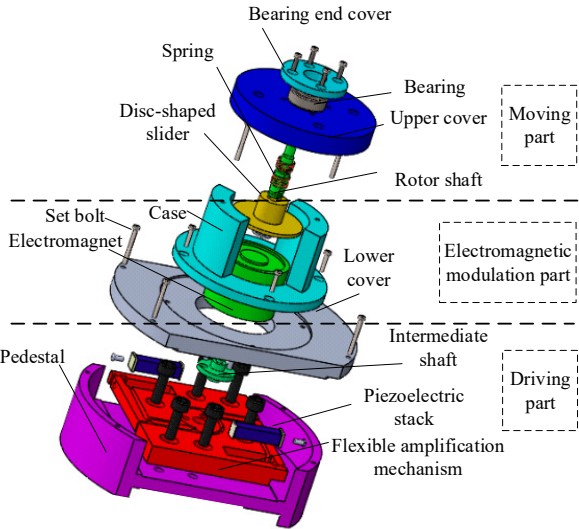

**Figure 1.** Motor structure.

In one part, two piezoelectric stacks cooperate with the flexible amplification mechanism for outputting swing motion, and then an intermediate shaft is assembled on the amplification mechanism as the driving part of the motor. In another part, the electromagnet matched with intermediate shaft, spring, and disc-shaped slider is combined as the electromagnetic modulation part of the motor. Finally, the drive part, the electromagnetic modulation part, and the moving part are assembled through the shell, end cover, pedestal, etc. to form a complete motor.

## 2.2. Working Principle

In the working step, the electromagnet attracts the disc-shaped slider and then rotates with the driving part. Considering that it takes a certain time for the disc-shaped slider to be attracted down after the electromagnet is energized, the piezoelectric stacks and the electromagnetic modulation mechanism are driven by two square wave signals with the same frequency but different phases separately. From Figure 2, $U_a$ is the peak value of the driving voltage for the electromagnetic modulation part, while $U_b$ is the peak value of the driving voltage for the piezoelectric stacks, $T$ is one cycle of the motor operation, and $\Delta t$ is the driving time difference of the two driving signals. This type of piezoelectric motor will present a stepping rotary motion, as shown in Figure 3.

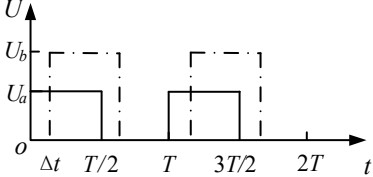

**Figure 2.** Drive signal.

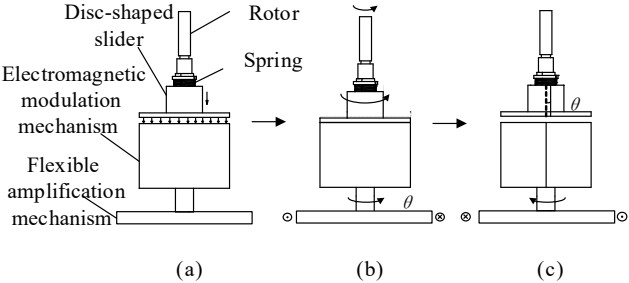

**Figure 3.** Working principle. (**a**) Attracting state; (**b**) Clamping and rotating; (**c**) Restoring state.

The details of the working principle are as follows:

- A square wave signal with peak value $U_a$ of voltage is supplied to the electromagnetic modulation mechanism. So, the disc-shaped slider is attracted under the action of the electromagnetic force and keeps in contact with the upper surface of the electromagnet, as shown in Figure 3a.
- After the time $\Delta t$, another square wave signal with peak value $U_a$ of voltage, as shown in Figure 2, is applied to the piezoelectric stacks. Two piezoelectric stacks will produce elongation deformation, and then drive the electromagnetic modulation part and the rotor shaft rotation angle $\theta$ through the flexible amplification mechanism, as shown in Figure 3b.
- When the high level of the driving signal $U_a$ disappears, as shown in Figure 2, the disc-shaped slider would be separated from the electromagnet under the action of the spring force. After another period of time $\Delta t$, the driving signal $U_b$ of the piezoelectric stacks is at low level. Then, two piezoelectric stacks restore to their original length and the electromagnetic modulation part returns to its original position with the elastic restoring force of the flexible amplifying mechanism, so the rotor rotates an angle $\theta$ relative to the electromagnet, as shown in Figure 3c. The time different $\Delta t$ of each cycle of two driving signals prevents that the rotor shaft reverses due to inertia when the disc-shaped slider is separated from the electromagnet. Repeating the above steps can achieve continuous step rotation of the motor.

## 3. Analysis for Free Vibration

For solving the natural frequency and modal function, first, we need to establish the free vibration dynamic equation of the proposed piezoelectric motor. The key part of the piezoelectric motor mainly includes a driving part, an electromagnetic modulation part, and a moving part. For the establishment of the free vibration dynamics model, we consider the system condition of clamping the rotor with the electromagnetic force. Therefore, the electromagnetic modulation part and the moving part are simplified into one mass. Meanwhile, the flexible amplification mechanism also is simplified, as shown in Figure 4. According to the different vibration and structural characteristics of each component, free vibration equations of each subsystem need to be established respectively, and then the boundary condition and continuous condition can be utilized to solve the whole system free vibration characteristics. Since the structure of the flexible amplification mechanism is center symmetrical, here, only one side is used to carry out calculations.

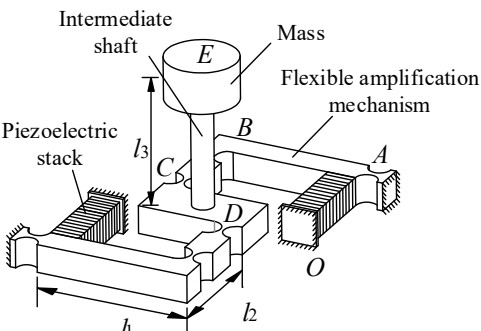

**Figure 4.** Simplified model of piezoelectric system for free vibration.

### 3.1. Free Vibration Modeling of a Piezoelectric Stack

The piezoelectric stack elongates or recovers in the axial direction under signal excitation, so it can be simplified into a continuous system of axial vibration. The piezoelectric stack dynamics model is shown in Figure 5, in which $\delta(y, t)$ is the axial displacement equation of the piezoelectric stack, $F$ is the axial internal force between the piezoelectric sheets, and $f$ is the axial distribution force. Let the density of the piezoelectric material be $\rho_p(y)$, the elastic modulus is $E_p(y) = c_{33}$, and the cross-sectional area is $A_p$.

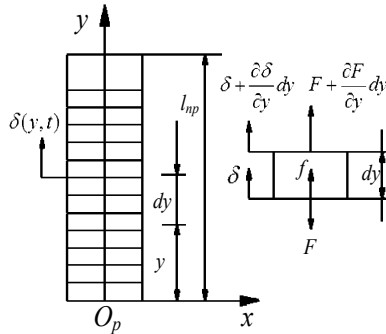

**Figure 5.** Free vibration model of piezoelectric stack.

According to the dynamic model, the dynamic differential equation of the piezoelectric stack is established as follows:

$$\rho_p A_p \frac{\partial^2 \delta}{\partial t^2} = c_{33} A_p \frac{\partial^2 \delta}{\partial y^2} \tag{1}$$

Let the solution of Equation (1) be:

$$\delta(y, t) = \phi(y) q(t) \tag{2}$$

The modal function of piezoelectric stack can be obtained as:

$$\phi(y) = W_1 \sin \frac{\omega y}{a} + W_2 \cos \frac{\omega y}{a} \tag{3}$$

where $a = \sqrt{c_{33}/\rho_p}$, the coefficients $W_1$, $W_2$ and the natural frequency $\omega$ can be determined by the boundary conditions.

### 3.2. Free Vibration Modeling of Flexible Amplification Mechanism

Half of the flexible amplification mechanism is simplified into two beams (*AB* and *BC*) and a mass (*CD*). Two beams are considered as an elastomer and the mass is considered as rigid body. Each simplified unit is connected to others by flexible hinges. The dynamic model of half of the flexible amplification mechanism is shown in Figure 6. The flexible hinge is simplified into a combination of a rigid hinge and a coil spring. This part, the beam AB is taken as a research subject. We design the length of the beam *AB* to be greater than 5 times the height of the cross-sectional, so the driving beam *AB* is considered as an Euler–Bernoulli beam [10]. The origin of a coordinate system is set at point *A*. x-axis direction is along the length direction of the beam *AB*. When the beam *AB* is excited by piezoelectric stack, it presents bending vibration. Let the longitudinal displacement of the corresponding bending vibration of the beam be $y_1 (x_1, t)$, the longitudinal distribution force on the beam be $f_1$, the length of beam *AB* be $l_1$, the density of the beam be $\rho_q$, the elastic modulus be $E$, the cross-sectional area be $S_1$, and the longitudinal shear force be $F_s$.

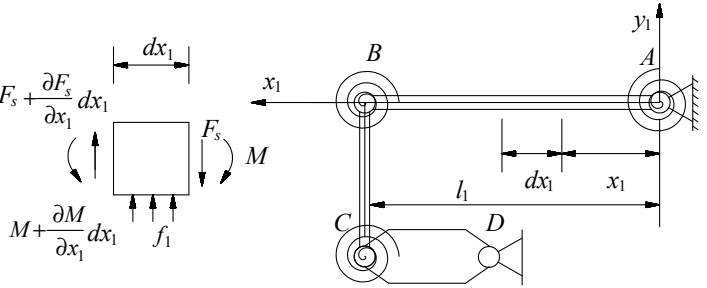

**Figure 6.** Free vibration model of driving beam.

According to the dynamic model, the dynamic differential equation of the beam *AB* is established as follows:

$$EI\frac{\partial^4 y_1(x_1,t)}{\partial x_1^4} + \rho_q S_1 \frac{\partial^2 y_1(x_1,t)}{\partial t^2} = 0 \tag{4}$$

Let the solution of Equation (4) be:

$$y_1(x_1,t) = \phi_1(x_1)q_1(t) \tag{5}$$

The modal function of the beam *AB* can be obtained as:

$$\phi_1(x_1) = C_1 \cos\beta x_1 + C_2 \sin\beta x_1 + C_3 \cosh\beta x_1 + C_4\sinh\beta x_1 \tag{6}$$

where $\beta = \sqrt[4]{\rho_q S_1 \omega / EI}$, the coefficients $C_1$–$C_4$ and natural the frequency $\omega$ can be determined by the boundary conditions.

As the piezoelectric stack forces the beam *AB* to bending vibrate, the beam *BC* presents longitudinal vibration and slight lateral vibration. Here, the slight lateral vibration is negligible. We assume the length of the beam *BC* is $l_2$, the cross-sectional area is $S_2$, and the density constant of the beam is $\rho_q$.

Point *B* is set as origin of a relative coordinate system, and $x_2$ axis is along the length direction of beam *BC*. As shown in Figure 7, $F_2$ is the axial internal force acting on the beam cross section, $f_2$ is the axial distribution force, $l_m$ is the length of the beam *CD*. Let the longitudinal vibration displacement of beam *BC* is $u(x_2,t)$.

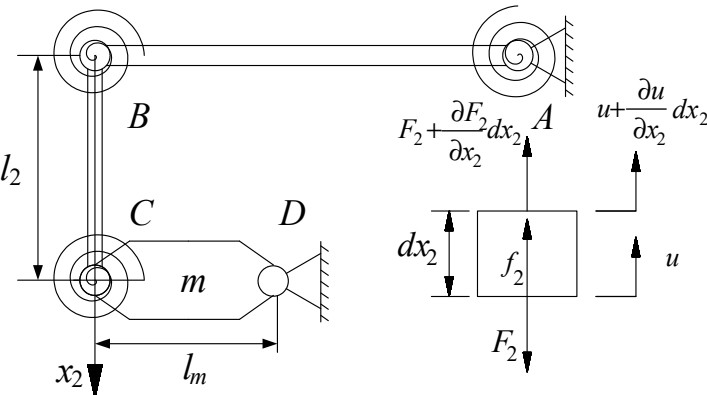

**Figure 7.** Free vibration model of intermediate beam mass.

According to the dynamic model, the dynamic differential equation of the beam *BC* is established as follows:

$$\frac{\partial^2 u_2}{\partial t^2} = \frac{E}{\rho_q}\frac{\partial^2 u_2}{\partial x_2^2} \tag{7}$$

Let the solution of Equation (7) be:

$$u_2(x_2,t) = \phi_2(x_2)q_2(t) \tag{8}$$

The modal function of beam *BC* can be obtained as:

$$\phi_2(x_2) = B_1 \sin\omega\sqrt{\frac{\rho_q}{E}}x_2 + B_2 \cos\omega\sqrt{\frac{\rho_q}{E}}x_2 \tag{9}$$

The coefficients $B_1$, $B_2$ and the natural frequency $\omega$ can be determined by the boundary conditions.

### 3.3. Free Vibration Modeling of the Intermediate Shaft

The intermediate shaft is utilized to connect the electromagnetic modulation mechanism and the flexible amplification mechanism. The subsystem vibration model of the intermediate shaft and the electromagnetic modulation mechanism is shown in Figure 8, the simplified mass is arranged at an end of the shaft. Considering the intermediate shaft *DE* to be an ideal elastic body, when the motor is running, the intermediate shaft presents torsional vibration.

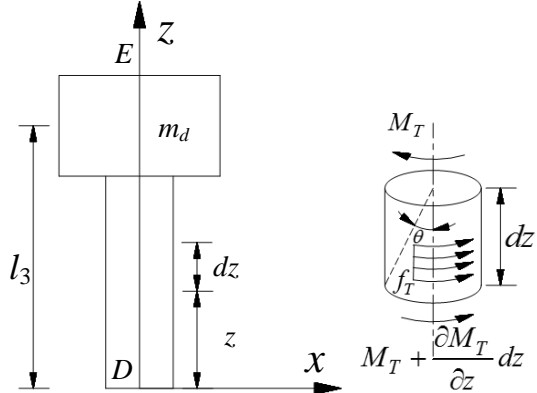

**Figure 8.** Free vibration model of intermediate shaft.

Assume that the material of the intermediate shaft is uniform, the density constant is $\rho_z$, the shear modulus is $G$, and the second moment of the cross section of the shaft is $I_z$. Point $D$ is set as the origin of a relative coordinate system, $z$ axis direction points to simplified mass $m_d$, $\theta$ is the angular displacement of the shaft torsional vibration, and $f_T$ is the torque per unit length in the axis direction.

According to the relationship between torque and angle, the partial differential equation of the torsional vibration of the intermediate shaft is:

$$\frac{\partial^2 \theta}{\partial t^2} = c^2 \frac{\partial^2 \theta}{\partial z^2} \tag{10}$$

Let the solution of Equation (10) be:

$$\theta(z,t) = \phi_3(z) q_3(t) \tag{11}$$

The modal function of the shaft DE can be obtained as:

$$\phi_3(z) = N_1 \cos\left(\frac{\omega z}{c}\right) + N_2 \sin\left(\frac{\omega z}{c}\right) \tag{12}$$

where $c = \sqrt{G/\rho_z}$, the coefficients $N_1$, $N_2$ and the natural frequency $\omega$ can be determined by the boundary conditions.

### 3.4. Boundary and Continuity Conditions

After each subsystem vibration mode function of the motor is established, for solving the natural frequencies and the coefficients of the mode function, the boundary conditions and continues conditions need to be discussed and the mechanical relationship between subsystems determined.

From Figures 5–8, the following relationship can be obtained at no load:

1.  The end $O_p$ of piezoelectric stack is the fixed, so the displacement is zero at $x = 0$. It can be expressed as:

$$\phi(0) = 0 \tag{13}$$

2.    Regarding the beam *AB*, the displacement at point *A* of the driving beam *AB* is zero, and the bending moment is equal to the reverse moment of the coil spring, so:

$$\begin{cases} \phi_1(0) = 0 \\ EI\phi_1''(0) = -k\phi_1'(0) \end{cases} \tag{14}$$

where *k* is the bending stiffness of the flexible hinge.

3.    At the thrust point of the piezoelectric stack, the displacement equals to the elongation of the piezoelectric stack, and the thrust at $y = l_{np}$ and the shear force at $x_1 = l_{Fp}$ are equal, so:

$$\begin{cases} \phi(l_{np}) = \phi_1(l_{F_p}) \\ c_{33}A_p\phi'(l_{np}) = EI\phi_1'''(l_{F_p}) \end{cases} \tag{15}$$

4.    At endpoint *B* of beam *AB*, since the flexible hinge provides an elastic resilience, the shear force of point *B* equals the elastic force of the flexible hinge. It can be expressed as:

$$EI\phi_1''(l_1) = -k\phi_1'(l_1) \tag{16}$$

5.    Regarding the beam *BC*, when the flexible amplification mechanism is operating, beam *BC* exist a translational motion, and let the translational displacement be $x_p$. Therefore, the displacement of the endpoint *B* of the beam *AB* is equal to the sum of the displacement of the endpoint *C* of the beam *BC* and the translational displacement $x_p$. In addition, the shear force at $x_1=l_1$ and the thrust at $x_2 = 0$ are equal, so:

$$\begin{cases} \phi_1(l_1) = \phi_2(0) + x_p \\ EI\phi_1'''(l_1) = ES_2\phi_2'(0) \end{cases} \tag{17}$$

6.    At the endpoint *C* of beam *BC*, the superposition of the torque generated by the translation of the beam *BC* and the end moment of the bending beam *AB* are equal to the reverse moment of the flexible hinge, so:

$$ES_2\phi_2'(l_2)l_m + EI\phi_1''(l_1) = -k\phi_2(l_2) \tag{18}$$

7.    For the junction of the intermediate shaft and mass *m*, the sum of the displacement at point *C* and the translational displacement $x_p$ is equal to the torsional angular displacement times the length $l_m$. The torque at the endpoint *E* of the intermediate shaft is equal to the reverse torque of the mass $m_d$, so the boundary condition of the torsional intermediate shaft is:

$$\begin{cases} \phi_2(l_2) + x_p = l_m\phi_3(0) \\ GI_p\phi_3'(l_3) = -J\omega_3^2\phi_3(l_3) \end{cases} \tag{19}$$

Then, substitute Equations (13)–(19) into the mode functions, respectively, to solve the natural frequency of each order of the system and the component mode coefficients.

## 4. Result Analysis

The flexible amplification mechanism was made of AA 7075 aluminum alloy. The intermediate shaft was made of AISI 1045 steel. The type of the piezoelectric stack was PSt150/5 × 5/20 (Core Tomorrow Technology Co., Ltd., Harbin, China). Table 1 shows the material properties and geometric parameters of piezoelectric stacks, flexible amplification mechanism and mass.

**Table 1.** Free vibration calculation parameters.

| $\rho_p$ (g/cm$^3$) | $\rho_q$ (g/cm$^3$) | $I_q$ (cm$^4$) | $m_d$ (kg) | $S_1$ (mm$^2$) |
|---|---|---|---|---|
| 7.5 | 2.81 | 52.08 | $3.58 \times 10^{-3}$ | 25 |
| $S_2$ (mm$^2$) | $\rho_z$ (g/cm$^3$) | $L_{\mathrm{m}}$ (mm) | $G$ (Gpa) | $J$ (kg/m$^2$) |
| 20 | 7.85 | 10.77 | 80.7 | $2.11 \times 10^{-3}$ |
| $d_{33}$ (C/N) | $c_{33}$ (Gpa) | $l_{np}$ (mm) | $A_p$ (mm$^2$) | $b$ (mm) |
| $635 \times 10^{-12}$ | 55.6 | 20 | 25 | 5 |
| $l_{Fp}$ (mm) | $l_1$ (mm) | $l_2$ (mm) | $l_3$ (mm) | $E$ (Gpa) |
| 6 | 40.64 | 19.73 | 20 | 72 |

Under no-load state, substituting these parameters in Table 1 into Equations (13)–(19), the coefficients $W_1$, $W_2$, $C_1$–$C_4$, $B_1$, $B_2$, $N_1$, $N_2$ and the natural frequencies of the piezoelectric stack, beams and intermediate shaft are obtained. The first three order natural frequencies are shown in Table 2.

**Table 2.** First three order natural frequency under no-load state.

| Order | Natural Angular Frequency (rad/s) | Natural Frequency (Hz) |
|---|---|---|
| 1 | 291 | 46 |
| 2 | 227,207 | 36,179 |
| 3 | 503,641 | 80,198 |

According to the first three natural frequencies and coefficients $W_1$, $W_2$, $C_1$–$C_4$, $B_1$, $B_2$, $N_1$, $N_2$, the modal function curves of the piezoelectric stack, beams of flexible amplification mechanism and intermediate shaft under no-load state are drawn by MATLAB, as shown in Figures 9–11.

From Table 2 and Figures 9–11, the following points can be summarized:

- The natural frequencies of the 2nd and 3rd orders are greatly increased comparing with the 1st order. This is because when the 1st order free vibration is excited, the deflection of the piezoelectric stack, beam *AB*, and intermediate shaft *DE* basically present a linear change, and the deformation of the flexible amplification mechanism mainly occurs at the hinge, as shown in Figure 9. In addition, the modal stiffness of the flexure hinge is much smaller than the other beams, so the 1st order natural frequency is much lower than high order natural frequencies.

- With the increase of the order, the modal functions of each component begin to present bending deformation. In the 2nd order modal functions curve, the beam *AB* appears a peak near 0.01 m and a trough near 0.027 m. In the 3rd order modal function curve, the piezoelectric stack appears a peak near 0.008 m, the beam *AB* appears a trough near 0.023 m and two peaks near 0.012 m and 0.036 m, and the shaft *DE* appears a trough near 0.01 m. However the beam *BC* has basically no obvious peaks and troughs.

- Taking into account that the electromagnetic modulation mechanism is not suitable for working at high frequency, when the working frequency is selected to operate the motor, it should be less than the 1st order natural frequency.

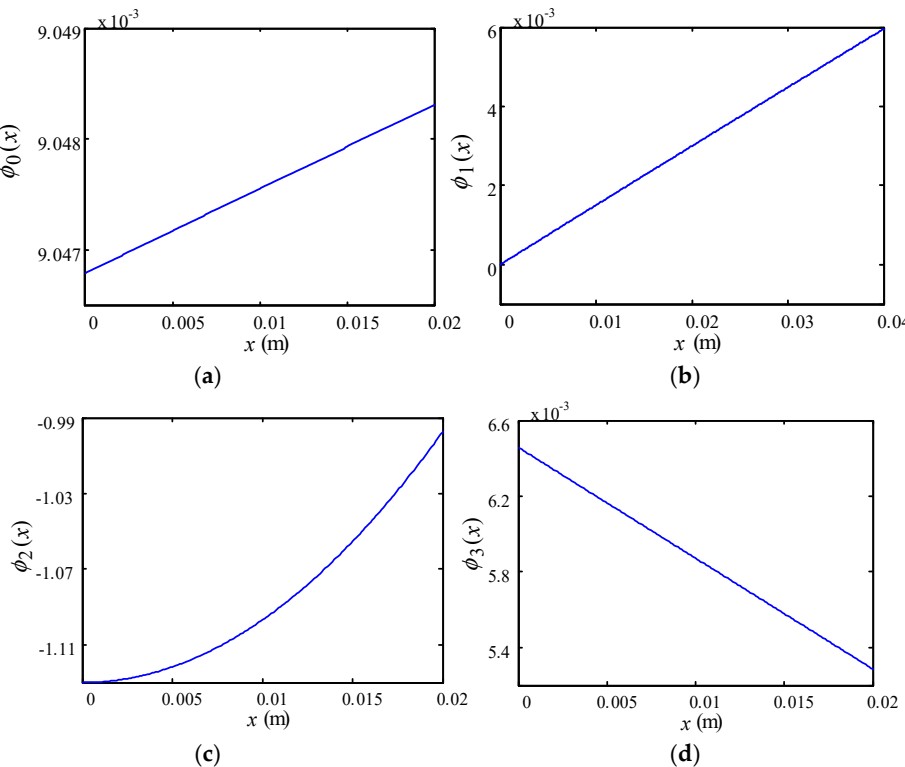

**Figure 9.** The 1st order modal function. (**a**) Piezoelectric stack; (**b**) beam *AB*; (**c**) beam *BC*; (**d**) intermediate shaft *DE*.

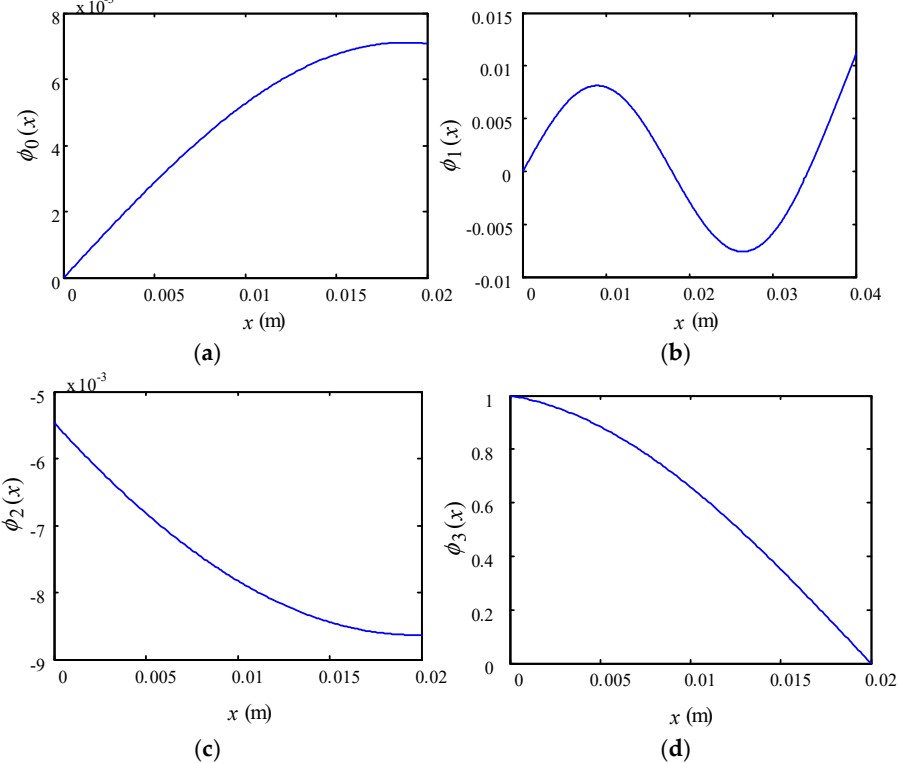

**Figure 10.** The 2nd order modal function. (**a**) Piezoelectric stack; (**b**) beam *AB*; (**c**) beam *BC*; (**d**) intermediate shaft *DE*.

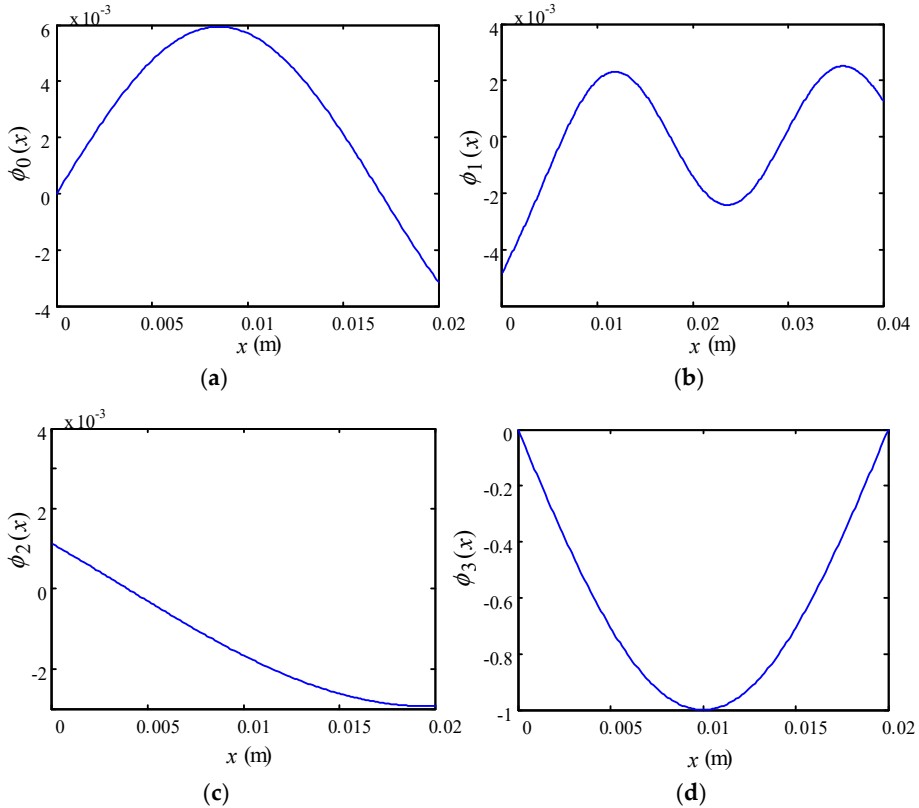

**Figure 11.** The 3rd order modal function. (**a**) Piezoelectric stack; (**b**) beam *AB*; (**c**) beam *BC*; (**d**) intermediate shaft *DE*.

## 5. Experiment and Discussion

To verify the analysis of free vibration of the piezoelectric motor, the test of the flexible amplification mechanism under excitation of the piezoelectric stacks was conducted. Then, the output characteristics of the prototype were measured for speed and torque.

### 5.1. Test for Free Vibration

In the free vibration experiment, a test system was built. The equipment included a computer, a HPV piezo drive power (Suzhou Boshi Robotics Technology Co., Ltd., Suzhou, China), the driving part of the motor, a OptoMET Digital LDV Vector Series laser vibrometer (OptoMET, Darmstadt, Hesse, Germany), and IOtech 625u data acquisition equipment (Beijing Perfect Technology Co., Ltd., Beijing, China). Figure 12 shows the flow chart of the free vibration test system. Figure 13 shows the experimental environment for measuring free vibration.

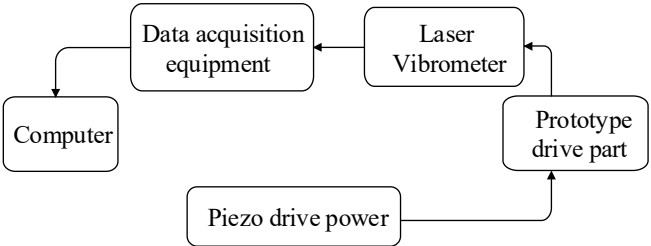

**Figure 12.** Flow chart of the free vibration test system.

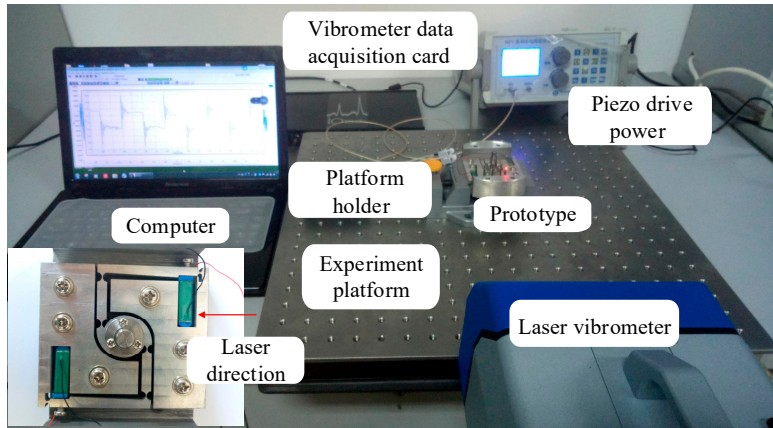

**Figure 13.** The experimental environment for measuring the free vibration.

After establishing the test system, 5 test points with equal spacing were selected on the beam *AB*, as shown in Figure 14. Due to the limitation of experimental conditions, beam *AB* among the beams of the flexible amplification mechanism was easier to measure than other beams, and could fulfil the requirements of verifying free vibration theoretical calculations, so it was used as the test object. On the beam *AB*, the spacing of the five points was chosen by 8.5 mm. In addition, the theoretically calculated result of 1st order natural frequency was about 46.2 Hz, so the excitation frequency of 0~100 Hz was used for the frequency sweep test. In order to improve test accuracy and precision, each test frequency was recorded and averaged after multiple excitations.

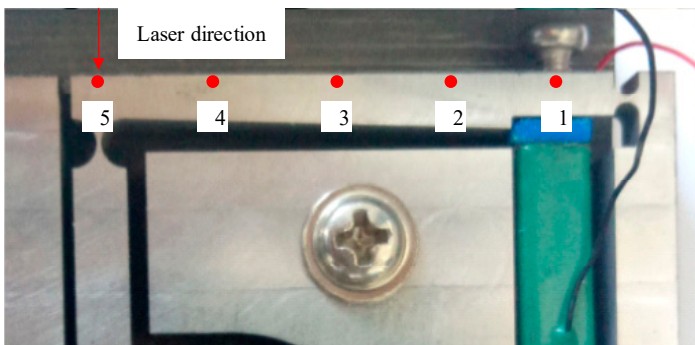

**Figure 14.** The beam *AB* free vibration test.

For visual analysis of experimental and theoretical values, the frequency domain diagram of the 5th point on the beam *AB* was drawn and using the data of 5 test points, the mode shape was also fitted by the curve, as shown in Figure 15.

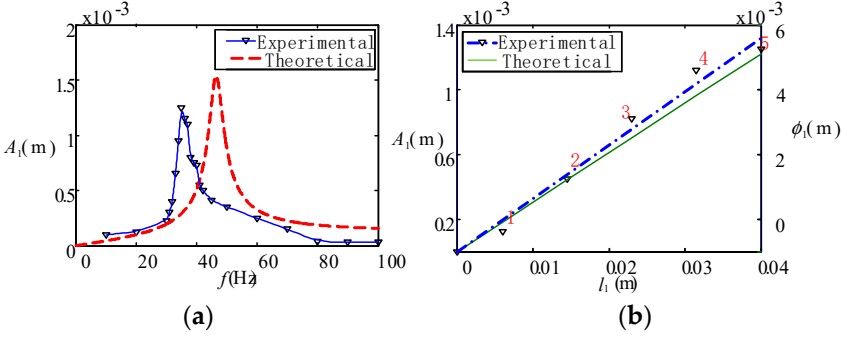

**Figure 15.** The 1st order modal experimental results of beam *AB*. (**a**) Frequency domain diagram of the 5th point on the beam *AB*; (**b**) mode shape.

From Figure 15, the following conclusions can be obtained:

- As shown in Figure 15a, for the 1st order natural frequency, the measured value is 38.5 Hz which is slightly lower than the theoretical value of 46.3 Hz. For the resonance displacement response amplitude of the beam *AB* end, the measured value is 1.35 mm which also is slightly lower than the theoretical value of 1.55 mm. The errors are 16.3% of the 1st order natural frequency and 12.9% of the response displacement.
- As shown in Figure 15b, the displacement response of the 5 points marked on the beam *AB* conforms to the theoretically calculated 1st order free vibration. The resonance response displacement of 5th point is the largest, about 1.35 mm. The 1st point is located at the junction of the driving beam and the piezoelectric stack, and the resonance response displacement is the smallest, about 0.04 mm.
- For the aforementioned errors, two reasons are summarized. The first reason is there was an error in the simplified model. The second reason is there also will be errors in measurement and data processing of the experiment. In addition, a wear-resistant gasket was added between the piezoelectric stack and beam *AB*, which will affect the experimental results by changing the contact stiffness.

### 5.2. Test of Output Characteristics

For the measuring output characteristics of the piezoelectric motor, another test system was established, which mainly included the computer, NI USB-6009 data acquisition card (Shenzhen Boruitu Electronic Technology Co., Ltd., Shenzhen, China), HPV piezo drive power (Suzhou Boshi Robotics Technology Co., Ltd., Suzhou, China), prototype, UNI-T direct current (DC) power supply (Unitech (China) Co., Ltd., Dongguan, China) and HP-5 micro force gauge (Yueqing Aidebao Instrument Co., Ltd., Yueqing, China ). Figure 16 shows the flow chart of the output characteristic test system. Figure 17 shows the experimental environment for measuring output characteristics.

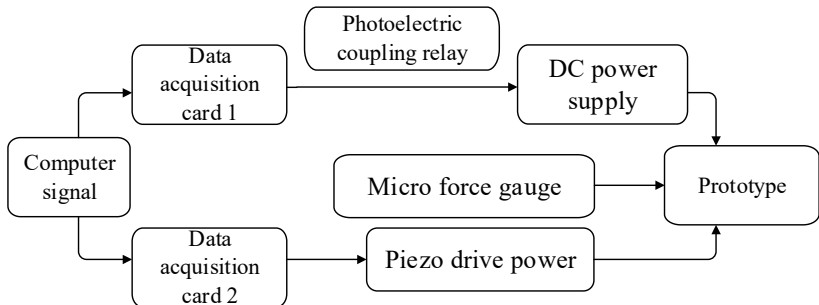

**Figure 16.** Flow chart of the output characteristic test system.

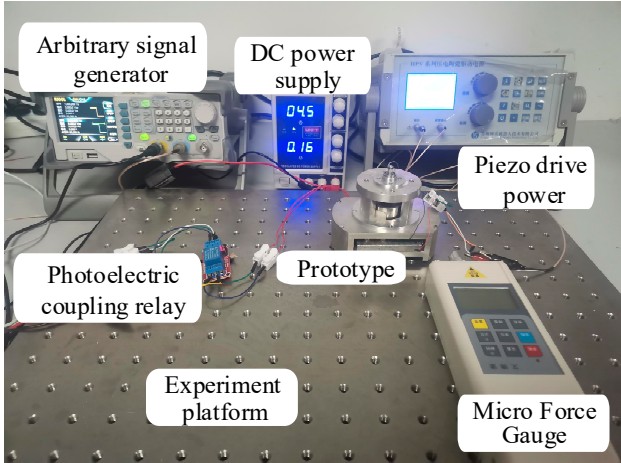

**Figure 17.** The experimental environment for measuring output characteristics.

In the test, the adjustment range of the drive voltage peak of the electromagnetic modulation mechanism was 3.5–5 V, the adjustment range of the drive signals frequency of the electromagnetic modulation mechanism and the piezoelectric stacks were 1–5 Hz, and adjustment range of the phase difference $\alpha$ between the electromagnetic modulation signal and the piezoelectric drive signal was 0–90°. In the speed measurement, the time of the prototype rotating a certain angle was recorded. Multiple measurements for speed were repeated to record the data, and the average speed of the prototype can be calculated. In the prototype torque measurement, a metal rod was mounted horizontally on the rotor shaft, and the thrust of the metal rod was measured with a micro force gauge. The micro force gauge could display the thrust of the metal rod test point, and the output torque could be calculated by the thrust. When the phase difference $\alpha$ was 50°, we investigated the operating status of the motor, as shown in Figure 18. The motor rotated about 9° in 120 s. This kind of motor presents a slow running state, from the measurement results.

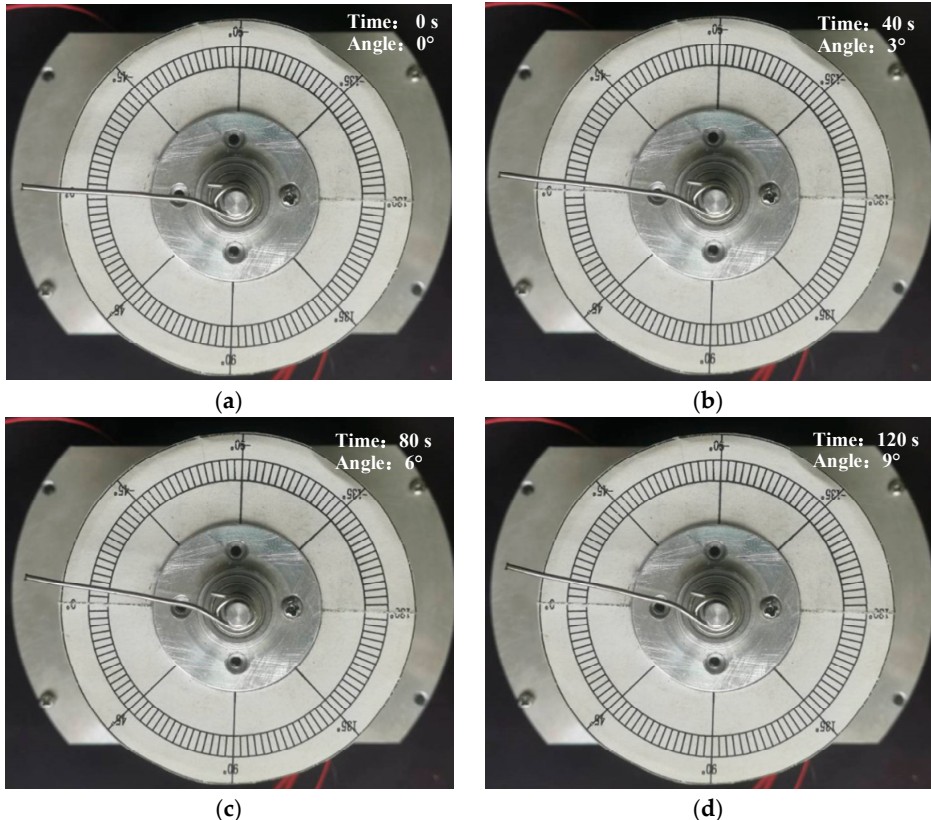

**Figure 18.** Experimental effect when the phase difference $\alpha$ is 50°. (**a**) At 0 s; (**b**) at 40 s; (**c**) at 80 s; (**d**) at 120 s.

To investigate the effect of driving signal parameters on speed and torque, more experiments were conducted. The change relationship curve of the speed and torque with different drive parameters was drawn, as shown in Figure 19.

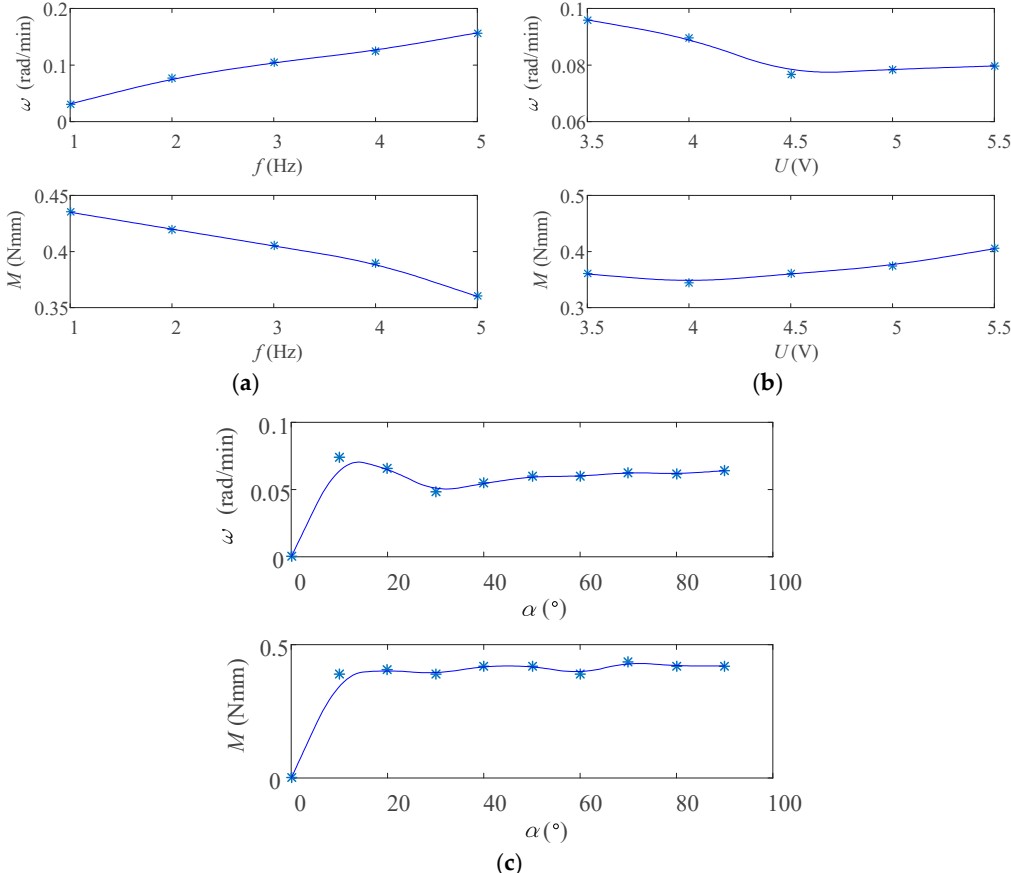

**Figure 19.** The law of the speed and torque of the prototype with different drive parameters. (**a**) Change with the driving frequency; (**b**) change with the electromagnetic drive voltage; (**c**) change with the phase difference $\alpha$ of the two drive signals.

From Figure 19, the following conclusions can be drawn:

- When the frequency of the driving signal increases, the motor speed gradually increases and the torque gradually decreases. Since the electromagnetic modulation mechanism sensitivity is lower than piezoelectric stack, the driving signal frequency is not suitable as it is too high. In order to maintain the good output performance of the motor, here we take the driving signal frequency of 3 Hz. This not only ensures the normal speed of the motor, but also enables the motor to have a strong load capacity.

- With the increase of the voltage of the electromagnetic driving signal, the motor speed decreases first and then tends to be stable. The overall torque shows a trend of slowly increasing. Since the electromagnetic voltage increases, the disc-shaped slider would be magnetized, which causes it to be unable to separate from electromagnet in time. Therefore, the electromagnetic driving voltage cannot easily exceed 4.5 V.

- When increasing the phase difference $\alpha$ between the two drive signals, the motor speed increases first and then tends to be stable, and the change law of the torque is similar to the speed. In order to ensure the normal operation of the motor, the drive signal needs to keep a phase difference. In this way, it can avoid the electromagnet and the disc-shaped slider being unable to separate in time during operation. It explains why there is a time difference $\Delta t$ between the two excitation signals in the driving principle.

## 6. Conclusions

A novel piezoelectric motor modulated by a magnetic field was proposed. The dynamic equations of the piezoelectric system were established for free vibration, the natural frequency and the vibration mode function were solved according to the continuity and boundary conditions, and the modal function curves were drawn and analyzed. In the free vibration test of the prototype, the relative error of the natural frequency between the test and the theoretical value was small, which verified the correctness of the free vibration theoretical analysis. In the output characteristics test, the prototype had a relatively stable output performance when the frequency of the motor drive signal was 3 Hz, the electromagnetic drive voltage was 4.5 V, and the phase difference between the electromagnetic modulation signal and the piezoelectric stack drive signal was 30°. This time, the motor output speed was 0.1046 rad/min, and the output torque was 0.405 Nmm. The novel electromagnetic modulation piezoelectric motor had some deficiencies. For example, due to the large coil volume, the motor was not easy to miniaturize, and the response of the electromagnetic modulation mechanism was slower. However, it had a higher resolution than the electromagnetically modulated piezoelectric motor with a speed of 2.28 rad/min (0.038 rad/s) proposed by Xing J. et al. [17]. The proposal of this novel piezoelectric motor provides a new idea for solving the friction problem of piezoelectric motors, and is of great significance for the further development and application of piezoelectric motors and dynamics research.

**Author Contributions:** Supervision, J.X.; project administration, J.X.; funding acquisition, J.X.; Conceptualization, J.X. and Y.Q.; methodology, J.X.; software, Y.Q.; validation, J.X. and Y.Q.; investigation, Y.Q.; resources, J.X.; data curation, Y.Q.; writing—original draft preparation, J.X. and Y.Q.; writing—review and editing, J.X. All authors have read and agreed to the published version of the manuscript.

**Funding:** This research was funded by National Natural Science Foundation of China, grant number 51605423 and Hebei Natural Science Foundation Project, grant number E2018203116.

**Conflicts of Interest:** The authors declare no conflict of interest.

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
