# Peer review of "A Novel Low-Frequency Piezoelectric Motor Modulated by an Electromagnetic Field"

_actuators, doi:10.3390/act9030085_

Round 1
Reviewer 1 Report
The results, Presented in the manuscript, are rather interesting and are worth publishing. However, some issues need to be explained and revised before publication. First of all, English in this manuscript needs to be checked. It has a lot of places existing spelling and grammatical errors. For example, in the abstract, “form trail” is difficult to understand, should it from trial (test)? On page 8, boundary condition 7, what is “mass md”? Also, some references are described not clearly, for example, ref.{14}. the language needs to be improved and seek professional help.
The title of section 3.2 and 3.4 does not match the content. Is this a mistake you expressed or want to express more? Please check the title of all the sections.
In Fig.8(c) The meaning of the symbol α is not given. And what is the unit of the Abscissa value?
From the measured data of the experiment, the motor speed is rather lower, I think publishing a few pictures in the manuscript is better.
Author Response
Dear reviewer,
Thank you very much for carefully reviewing the paper and giving a lot of valuable comments. Those comments are all very helpful for revising and improving our paper, as well as providing important guidance significant to our researchers. According to these comments given by you, we have carefully checked and revised the paper. The revised paper has replied to all the problems and hopes to satisfy your requirements. The revised portion is marked in red in the paper.
Please see to the attachment for the main corrections and responses to the reviewer’s comments.

Reviewer 2 Report
This paper studies a Piezoelectric Motor Modulated by Electromagnetic Field. Paper has a good flow but needs to be amended to be ready to be published in this journal. Some comments for the authors:
- The literature needs to be improved. The cited paper need to be critically reviewed. Not enough information is available in the manuscript. This should be followed by finding the gap and highlighting the aim.
- In 5.1 please include the high-resolution picture from the prototype and include the details of the equipment which are used. The brand and resolution of the equipment are needed.
- It seems this device just can work in low frequencies. If it is correct please include the low frequency in the title of the paper.
- In the conclusion part, the results need to be compared with existed literature to show the improvement.
Author Response

(The authors gave the same response as above.)

Reviewer 3 Report
The development and creation of new types of piezoelectric motors is an urgent task for modern technical applications. The material presented in the manuscript is of interest to Readers of the “Actuators”. However, the Reviewer believes that this manuscript may be accepted for publication when answering some questions and making minor additions.
Questions and comments to the text of the manuscript:
- Problems of analyzing the operation of oscillatory systems, which include piezoelectric components and sources of electromagnetic field, is quite common in the literature. An example is the monograph
Erturk A., Inman D. J. Piezoelectric Energy Harvesting. N.-Y.: Wiley, 2011. 392 p.
In this work, various piezoelectric devices controlled, among other things, by a magnetic field are analyzed. This work, although it belongs to the field of power generation systems, deserves to be mentioned in the text of the article. The authors may find it possible and useful to add some information from this source to the manuscript.
- The operation of piezoelectric transducers is extremely strongly influenced by the ambient temperature. According to the Reviewer, it is necessary to add to the manuscript of the article information about its impact on the operation of the proposed piezomotor (or the limitations of research on this issue).
3. The Authors point out the disadvantages of piezoelectric motors, which are controlled in the contact version. However, they do not disclose the disadvantages of the piezoelectric engine based on their proposed technical solution.
Author Response

(The authors gave the same response as above.)

Reviewer 4 Report
The proposed piezoelectric motor could be interesting for other readers involve in the area. However, before the publication of this work some minor corrections/modifications should be implemented.
Line 16: correct ==> “The” results show “that” ….
Better description of the last three lines of the abstract…
Recent reference for traveling wave motors: https://doi.org/10.3390/mi11050517
Line 48: correct ==> proposed “a” new …
Line 50: correct ==> the “inchworm” motor …
Line 82: correct ==> Motor “structure”…
Try to use the same word for “disk” or “disc”
Line 115: correct ==> in Figure 2 “the disc-shaped”…
Line 142: correct ==> “subsection” …
Line 160: correct ==> “subsection” …
Line 168: reference about this … “is considered as an Euler-…”
Line 265: reference for the piezoelectric stack
Line 319: reference for the laser vibrometer
Line 337: correct ==> “was drawn and using… “
Figure 18: Have you tried to increase the driving frequency above 5 Hz? Why these limits?
Author Response

(The authors gave the same response as above.)
